# HemN2 Regulates the Virulence of *Pseudomonas donghuensis* HYS through 7-Hydroxytropolone Synthesis and Oxidative Stress

**DOI:** 10.3390/biology13060373

**Published:** 2024-05-24

**Authors:** Yaqian Xiao, Wang Xiang, Xuerui Ma, Donghao Gao, Hasan Bayram, George H. Lorimer, Reza A. Ghiladi, Zhixiong Xie, Jun Wang

**Affiliations:** 1Cooperative Innovation Center of Industrial Fermentation, Ministry of Education & Hubei Province, Hubei University of Technology, Wuhan 430068, China; xiaoyaqian@hbut.edu.cn (Y.X.); xiangwang@hbut.edu.cn (W.X.); 2010511235@hbut.edu.cn (X.M.); 2International Center for Redox Biology & Precision Medicine of Hubei Province, Hubei University of Technology, Wuhan 430068, China; 3Hubei Key Laboratory of Cell Homeostasis, College of Life Sciences, Wuhan University, Wuhan 430072, China; 2021202040036@whu.edu.cn; 4Department of Pulmonary Medicine, School of Medicine, Koc University, 34010 Istanbul, Turkey; habayram@ku.edu.tr; 5Department of Chemistry, University of Maryland, College Park, MD 20742, USA; glorimer@umd.edu; 6Department of Chemistry, North Carolina State University, Raleigh, NC 27695, USA; reza_ghiladi@ncsu.edu

**Keywords:** *Pseudomonas donghuensis* HYS, HemN2, 7-hydroxytropolone, GacS, oxidative stress

## Abstract

**Simple Summary:**

*Pseudomonas donghuensis* HYS has lethal virulence towards *Caenorhabditis elegans*. Anaerobic coproporphyrinogen III oxidase (HemN) is involved in *Pseudomonas* heme synthesis. However, no research thus far has examined the contribution of HemN to the virulence of *Pseudomonas*. There are four *hemN* genes in *P. donghuensis* HYS, and we reported for the first time that the deletion of the *hemN2* gene significantly reduced the virulence of *P. donghuensis* HYS towards *C*. *elegans*. HemN2 was negatively regulated by the Gac system and regulated bacterial virulence via 7-hydroxytropolone (7-HT) synthesis and redox levels. Our findings revealed the key role of HemN2 in bacterial virulence, which may help us to better understand the strong pathogenicity of the genus *Pseudomonas*.

**Abstract:**

Compared to pathogens *Pseudomonas aeruginosa* and *P. putida*, *P. donghuensis* HYS has stronger virulence towards *Caenorhabditis elegans*. However, the underlying mechanisms haven’t been fully understood. The heme synthesis system is essential for *Pseudomonas* virulence, and former studies of HemN have focused on the synthesis of heme, while the relationship between HemN and *Pseudomonas* virulence were barely pursued. In this study, we hypothesized that *hemN2* deficiency affected 7-hydroxytropolone (7-HT) biosynthesis and redox levels, thereby reducing bacterial virulence. There are four *hemN* genes in *P. donghuensis* HYS, and we reported for the first time that deletion of *hemN2* significantly reduced the virulence of HYS towards *C. elegans*, whereas the reduction in virulence by the other three genes was not significant. Interestingly, *hemN2* deletion significantly reduced colonization of *P. donghuensis* HYS in the gut of *C. elegans*. Further studies showed that HemN2 was regulated by GacS and participated in the virulence of *P. donghuensis* HYS towards *C. elegans* by mediating the synthesis of the virulence factor 7-HT. In addition, HemN2 and GacS regulated the virulence of *P. donghuensis* HYS by affecting antioxidant capacity and nitrative stress. In short, the findings that HemN2 was regulated by the Gac system and that it was involved in bacterial virulence via regulating 7-HT synthesis and redox levels were reported for the first time. These insights may enlighten further understanding of HemN-based virulence in the genus *Pseudomonas*.

## 1. Introduction

*Pseudomonas* is a gram-negative bacterium that is widespread in nature. This genus contains many pathogen species, including *P. aeruginosa*, *P. putida*, *P. fluorescens*, and *P. syringae* [1,2,3]. *P. donghuensis* is a new species of *Pseudomonas*, which exhibits antifungal and antibacterial activities [4,5]. *P. donghuensis* HYS is the model strain of *P. donghuensis* and therefore has become the representative of the entire species for etiological studies [6]. In slow-killing assays, the HYS strain has shown stronger virulence to *C. elegans* than human pathogenic bacteria *P. aeruginosa* PAO1 and PA14 and fish pathogen *P. putida* [7,8,9].

*P. donghuensis* HYS secretes nonfluorescent iron carrier 7-hydroxytropolone (7-HT) and fluorescent pyoverdine. 7-HT is involved in bacterial virulence as a secondary toxic substance [10,11]. Pyoverdine is an important virulence factor in bacteria, and *pvdA* is a key gene in the pyoverdine synthesis pathway. Studies have shown that pantothenic acid metabolism in HYS affects bacterial virulence by affecting the synthesis of 7-HT and pyoverdine [11]. The global GacS/GacA regulatory system may control iron uptake by regulating the production of iron scavenger, allowing bacteria to adapt to different iron environments. GacS/GacA inactivation has been shown to lead to upregulation of pyoverdine and downregulation of 7-HT [12].

The heme system exists in both gram-positive and gram-negative bacteria; the heme synthesis system is essential for bacterial toxicity. The synthesis of heme in bacteria is controlled by oxygen, iron, and heme content. Anaerobic coproporphyrinogen III oxidase (HemN) plays an important catalytic role in heme synthesis. HemN binds two S-adenosine-_L_-methionine (SAM) molecules at the active site and catalyzes the oxidative decarboxylation of the two propionic acid side chains of ring A and ring B of coproporphyrinogen III to the corresponding vinyl group to produce protoporphyrinogen IX [13,14]. *P. donghuensis* HYS contains four *hemN* genes, which are called *hemN1*, *hemN2*, *hemN3*, and *hemN4*, and all encode anaerobic coproporphyrinogen III oxidase. There are two *hemN* genes in the slow-growing *Rhizobium* of soybean, both of which encode anaerobic coproporphyrinogen III oxidase. HemN1 has been shown to be nonfunctional, whereas HemN2 plays a key role in the biosynthesis of heme under anaerobic conditions [15].

Changes in iron or heme levels can lead to redox imbalances [16]. The imbalance of redox homeostasis in organisms manifests as oxidative stress or reductive stress. Oxidative stress can lead to the mass production of reactive oxygen species (ROS) and reactive nitrogen species (RNS) in the body, causing oxidative damage to DNA, RNA, proteins, and membrane components, as well as apoptosis [17,18,19,20]. The primary source of RNS is the nitric oxide radical (•NO), which can be produced through the nitrate–nitrite–•NO pathway [21]. Nitrite and nitrate are two endogenous oxides of nitrogen that are toxic in the body, and ONOO^−^ is also a potent oxidant that can cause oxidative stress [22,23]. Excess ROS produce oxygen molecules, and •NO is oxidized by superoxide to generate peroxynitrite (ONOO^−^) and then nitrate, nitrate is converted to nitrite by nitrate reductase, and nitrite is broken down by nitrite reductase to form •NO [21].

Reductive stress levels can be measured by total antioxidant capacity (T-AOC) [24]. The role of redox in the virulence mechanism of *P. aeruginosa* has been reported [1,2,25]. Phenazines secreted by *P. aeruginosa* induce oxidative stress and exert their virulence effects in *C. elegans* [25,26]. The *P. aeruginosa* virulence-related quorum sensing system controls genes necessary for oxidative stress [27]. In *P. aeruginosa*, the virulence gene sigma-E regulates genes involved in resistance to heat and oxidative stress [28]. The expression of the Pht cluster genes within the *P. syringae* virulence genes is regulated by oxidative stress [3]. *Bacillus anthracis* use their own nitric oxide (NO) as a key defense against the oxidative burst of the immune system [29]. Bacterial nitric-oxide synthase (bNOS) contributes to methicillin-resistant *Staphylococcus aureus* (MRSA) innate immune and antibiotic resistance [30]; bNOS inhibitors can work synergistically with oxidative stress to enhance MRSA killing [31]. Bacterial synthesis of NO contributes to bacterial virulence, increasing resistance to the immune system and against the killing effects of antibiotics. RNA-seq of *P. aeruginosa* PAO1 after treatment with the antimicrobial agent thymoquinone showed that the expression of antioxidant-related genes (catalase and glutathione peroxidase) were significantly suppressed, suggesting that PAO1 defends itself against oxidative damage by depleting antioxidant enzymes [32].

Research into the function of HemN has mainly focused on the synthesis of heme. However, the role of HemN in bacterial virulence to the host remains unclear. In this study, four *hemN* genes in strain HYS were knocked out for the first time, and a slow-killing assay was performed to analyze *hemN* genes’ involvement in the functional differentiation of bacterial virulence towards *C. elegans*. We further explored the correlation between the *hemN* gene and *P. donghuensis* HYS high-yielding nonfluorescent iron scavenger 7-HT and redox levels, and whether it is regulated by the virulent Gac system, Cbr/Crc system, pantothenic acid synthesis pathway, and fluorescent pyoverdine pathway to elucidate the effect of *hemN* on the virulence of *Pseudomonas* towards *C. elegans*.

## 2. Materials and Methods

### 2.1. Strains, C. elegans, Zebrafish, and Growth Conditions

The strains and plasmids used in this study are shown in Appendix A. *P. donghuensis* HYS and its gene knockout and complemented strains were cultured at 30 °C, and *E. coli* at 37 °C. If necessary, the following concentrations of antibiotics (Sinopharm, Shanghai, China) were added for strains of *P. donghuensis* HYS and its derivatives: gentamicin 50 μg/mL, chloramphenicol 25 μg/mL, and kanamycin 50 μg/mL. For *E. coli*, gentamicin 10 μg/mL, and ampicillin at 100 μg/mL were added.

The *C. elegans* wild type Bristol N2 (*C. elegans* N2) were purchased from the Caenorhabditis Genetic Center and cultured on nematode growth medium (NGM) plates at 22 °C. The worms were treated with bleach, after which the progeny from the synchronous stage were collected for subsequent experiments. In its natural habitat, *C. elegans* feeds on micro-organisms (mainly bacteria), but this is difficult due to environmental constraints resulting in very slow growth. *E. coli* OP50 is a uracil nutrient-deficient phenotype, which can only obtain uracil from NGM medium and has restricted growth [33]. *E. coli* OP50 can provide *C. elegans* food without overgrowing under laboratory conditions, and is therefore widely used as a control strain in bacterial-host interactions studies using *C. elegans* as an animal model [33,34,35].

Zebrafish (AB series) were raised and maintained according to the standard procedure (http://www.zfish.cn/). Zebrafish were purchased from the China zebrafish resource center, maintained at 28 ± 0.5 °C, and photoperiod of 14 h of darkness and 10 h of light. The day before mating, zebrafish were placed in a breeding tank with a 2/1 ratio of males to females and separated by a transparent acrylic partition. The next morning from 7 a.m. to 8 a.m., the partition was removed and the zebrafish were freely fertilized. Fish eggs were collected in sterile petri dishes and washed and soaked with 0.001% methylene blue solution to prevent fungal infection.

### 2.2. DNA Manipulation and Plasmid Construction

Gene knockout strains were constructed using the homologous recombination principle. The upstream and downstream homologous arm segments of the target gene were amplified via PCR using specific primers, which were then separated by agarose gel electrophoresis and digested with primer-specific restriction enzymes (Thermo Fisher Scientific, Waltham, MA, USA). The resulting fragment was then ligated with T4 DNA Ligase to the suicide plasmid pEX18Gm and transformed into *E. coli* S17-1 λpir [36]. The correct knockout vector was identified by sequencing. Next, wild-type HYS was conjugated with the knockout vector, and the pEX18Gm was removed using a 10% (wt/vol) sucrose Luria broth (LB) plate. Mutant strains with the correct sequencing were stored at −80 °C for later use.

The complete gene sequence to be expressed was amplified, restriction enzyme digested and then linked to the broad-spectrum host vector plasmid pBBR1MCS-2 [37], transformed into *E. coli* S17-1(λpir), and conjugated with HYS knockout strains to construct complementation mutants [7]. M13 universal primer was used to identify the correct complemented strains of the target gene. The primers used in this study are shown in Appendix A.

### 2.3. Slow-Killing Assay

*E. coli* OP50 and HYS strains were inoculated into LB liquid medium and cultured at 37 °C and 30 °C, respectively, for 12 h under 200 rpm oscillation. A total of 200 µL of bacterial culture was added to the center of the NGM plates and incubated for 12 h at 22 °C in a constant-temperature incubator [7,38]. Twenty L4 stage *C. elegans* cultured simultaneously were selected and placed on the NGM experimental strains plates with 5 replicates per group (100 *C. elegans* were fed per strain). *E. coli* OP50 was used as a negative control, and wild-type HYS was used as a positive control. To eliminate the effects of reproduction, the surviving *C. elegans* were transferred daily to fresh plates of NGM experimental strains until all *C. elegans* died. The survival function curve of *C. elegans* was plotted using SPSS 18.0 Kaplan–Meier and the LT_50_ value was obtained. The LT_50_ value was plotted with Origin 9.0 software (OriginLab, Northampton, MA, USA). The experiment was repeated independently three times.

### 2.4. Growth Curve Analysis

Single colonies of *P. donghuensis* HYS and mutant strains were inoculated into LB liquid medium, and cultured at 200 rpm and 30 °C for 12 h. The 600 nm (*OD*_600_) of bacterial culture was adjusted to the same value. Then, 200 μL bacterial cultures were added to the NGM plates and incubated at 22 °C. The optical density at *OD*_600_ was measured using a V-1200 spectrophotometer (Mapada, Shanghai, China) at 14 consecutive time points. Three replicates were set for each strain at each time point and bacterial growth curves were plotted using Origin 9.0.

### 2.5. Statistics of Pharyngeal Pump Rate for C. elegans

*C. elegans* cultured at the L4 stage were selected and placed on growth medium (NGM) plates with experimental strains, and the frequency of the pharyngeal of *C. elegans* pump was measured under the microscope (SZM-45B1, Sunny Optical, Yuyao, China). Pharyngeal pumping rates were counted for 10 *C. elegans* in each experiment.

### 2.6. Analysis of Bacterial Colonization of the C. elegans Gut

Vectors containing green fluorescent protein were electrotransferred into strain OP50, wild-type, and mutant strains for expression and identification [19]. *E. coli* OP50 was used as a negative control, and wild-type HYS was used as a positive control. The resulting bacterial strains were then fed to *C. elegans* for 24 h. *C. elegans* were washed repeatedly with M9 buffer, anesthetized with sodium azide, and photographed under a fluorescence microscope (BZ-X810, Keyence, Osaka, Japan) [39].

### 2.7. RNA Extraction and RT-qPCR

Taking 200 μL of bacterial culture in the middle of logarithmic growth, we added it dropwise to the NGM culture. After solidification, the bacterial moss was scraped in 1 mL TRIzol reagent (Ambion; Austin, TX, USA) to extract total RNA, which was reverse transcribed into cDNA as a template for RT-qPCR. The RT-PCR used the Prime-Script RT reagent Kit with gDNA Eraser (TaKaRa, Kusatsu, Japan). The gDNA Eraser was used to remove genomic DNA, and the reaction conditions were 42 °C, 5 min, 4 °C, 10 s. Then, 10 μL of total RNA was mixed with 4 μL of 5× PrimerScript Buffer, 1.0 µL of PrimerScript RT Enzyme Mix I, 1.0 µL of RT Prime Mix and 4.0 µL of RNase Free ddH_2_O. The reaction procedures were 37 °C, 15 min, 85 °C, 5 s and 4 °C for cDNA.

For qPCR, 5 μL of template cDNA was mixed with 10 μL of 2 × Taq Pro Universal SYBR qPCR Master Mix (Vazyme, Nanjing, China) and 0.4 μM forward and reverse primers in a final volume of 20 μL. The qPCR cycling conditions were as follows: initial denaturation at 95 °C for 30 s, followed by 40 cycles of denaturation (95 °C for 10 s, 60 °C for 30 s). The melting curves were acquired using the instrument default acquisition procedure. Fluorescence quantitative PCR reactions were run on a qPCR machine (CFX96 Real-Time System; Technologies; Hercules, CA, USA). Data were analyzed with the quantitative fluorescence analysis software Bio-Red CFX Manager and calculated using the 2^−ΔΔC^_T_ method [40].

### 2.8. Quantification of 7-HT Siderophore Production

*P. donghuensis* HYS produces nonfluorescent iron scavenger 7-HT with characteristic absorption peaks at approximately 330 nm and 392 nm, and fluorescent pyoverdine with characteristic absorption peaks at approximately 405 nm in liquid MKB [10,11]. Fresh single colonies were picked and inoculated into liquid modified King’s B (MKB) medium (the final pH was approximately 7.2, comprising 15 mL glycerol, 2.5 g K_2_HPO_4_, 5 g casamino acids and 2.5 g MgSO_4_ per liter), followed by incubation at 30 °C and at 200 rpm for 24 h. The bacterial cultures were normalized to *OD*_600_ = 0.25 with fresh pre-sterilized liquid MKB medium. The supernatant of bacteria culture was collected by centrifugation and filtration. Subsequently, the characteristic peaks for 7-HT were measured with full wavelength scanning of the UV-visible spectrophotometer (UV-2550, Shimadzu, Kyoto, Japan). The absorption spectra of the filtered supernatants were measured every 0.5 nm, using MKB liquid medium as a baseline.

### 2.9. Quantification of Total Antioxidant Capacity

Bacterial test solution preparation was as follows. Fresh single colonies of wild-type HYS and mutant strains were seeded in LB liquid medium and incubated at 30 °C for 12 h with shaking at 200 rpm, and the *OD*_600_ of the bacterial culture was adjusted to be consistent. The molecules were released from them by ultrasonic crushing using an ultrasonic cell crusher. The ultrasonication procedure was 2 s, 4 s pause, 6 min duration, and the ultrasonication power was 15%–20%. The supernatant obtained by centrifugation at 4 °C, 12,000 rpm, and 5 min was used as the bacterial detection solution.

The total antioxidant capacity of the bacterial solution was determined using the ABTS method, which is based on the oxidation of 2,2′-Azinobis-(3-ethylbenzthiazoline-6-sulphonate) (ABTS) diamine salts by potassium persulfate to produce green ABTS free radicals (ABTS•), which are eliminated in the presence of hydrogen-supplying antioxidants [24]. Take 5 μL bacterial solution and add it to 400 μL ABTS working solution. Incubate at room temperature for 6 min in the dark with a UV-visible spectrophotometer Cary 60 (Agilent, Santa Clara, CA, USA) at 200–900 nm. The standard curve was drawn based on the absorbance of the standard solution detected at 734 nm. The sample’s absorbance was measured and inputted into the standard curve to calculate the mixture’s antioxidant capacity.

### 2.10. Quantification of Nitrite and Nitrate

For the preparation of the bacterial test solution, refer to Section 2.9. Quantitative detection was performed using a Nitric Oxide Analyzer 280i (General Electrics, Boston, MA, USA) based on chemiluminescence analysis. The nitrite and nitrate in the bacterial solution reacted with the triiodide ion and vanadium trichloride in the sample to produce NO gas [21]. The peak area of the sample was plotted on the standard curve to calculate the concentration.

### 2.11. P. donghuensis HYS Immersion Infection of Zebrafish Larvae

Single colonies were selected and inoculated into LB liquid medium and grew in a 220 rpm shaker at 30 °C for 12 h. *OD*_600_ was determined by centrifugation and suspension of the fresh bacterial culture with PBS. The immersion method was used to carry out the infection experiment of zebrafish larvae [41]. The bacterial culture was diluted successively and added to 48-well plates, each containing 10 mL sterilized PBS per well. This ensured that the final concentration of *P. donghuensis* HYS bacterial suspension was 6 × 10^6^ CFU/mL, respectively. There were 10 zebrafish larvae in each 48-well plate, and the zebrafish larvae in ultra-pure water were used as the negative control. The zebrafish survival rate was measured every 2 h.

### 2.12. Statistical Analysis

The Kaplan–Meier survival function curve of *C. elegans* and zebrafish was plotted using IBM SPSS 18.0 (SPSS Inc., Chicago, IL, USA), and the LT_50_ values were also obtained and plotted using Origin 9.0 (OriginLab, Northampton, MA, USA). Comparisons between groups were performed using a two-tailed unpaired Student’s *t*-test with Prism 7 (GraphPad, San Diego, CA, USA). Data are presented as the mean ± SD from three independent experiments.

## 3. Results

### 3.1. HemN Participates in the Virulence of P. donghuensis HYS to C. elegans

*P. donghuensis* HYS contains four *hemN* genes, which were called *hemN1*, *hemN2*, *hemN3*, and *hemN4* according to the genomic sequence for ease of differentiation. All of these encode anaerobic coproporphyrinogen III oxidase. Multiple amino acid sequences revealed that the protein sequence similarity of HemN1, HemN2, HemN3, and HemN4 was 43.94% (Appendix A). The results of slow-killing assays showed that the median lethal time (LT_50_) for *C. elegans* exposed to wild-type HYS was 3.4 ± 0.08 days. The Δ*hemN2* strain had significantly reduced virulence (the LT_50_ increased to 5.5 ± 0.2 days) (Figure 1A,B). However, the knockout strains Δ*hemN1*, Δ*hemN3*, and Δ*hemN4* showed no significant reductions in virulence.

Furthermore, *C. elegans* exposed to Δ*hemN12* (LT_50_ 4.633 ± 0.135 days) and Δ*hemN124* (LT_50_ 5.178 ± 0.212 days) had the same trend of reduced virulence as Δ*hemN2* (Figure 1C,D). *hemN4* had a compensatory role in Δ*hemN12*. In addition, the virulence of the complemented strain Δ*hemN2*/pBBR2-*hemN2* was restored to wild-type levels, further confirming the critical role of *hemN2* in bacterial virulence (the LT_50_ value was 3.39 ± 0.102 days) (Figure 1E,F). These results showed that *hemN2* deletion significantly reduced the virulence of HYS towards *C. elegans*, whereas the other three genes did not show a reduction in virulence. In addition, *hemN4* played a compensatory role for the loss of *hemN1* and *hemN2*.

### 3.2. HemN2 Reduced the Colonization of P. donghuensis HYS in the C. elegans Gut

To confirm that the reduced virulence of Δ*hemN2* strains towards *C. elegans* was not due to a growth disadvantage resulting from gene deletion, we measured the growth curve of Δ*hemN2* on NGM plates. The growth curve of strain Δ*hemN2* showed similar growth trends and no obvious growth defects compared to the HYS strain (Figure 2A). This indicated that the reduced virulence of the Δ*hemN2* strain was indeed related to the gene itself and was not caused by insufficient bacterial growth.

*C. elegans* remains in a state of continuous feeding during its growth; the amount consumed depends on the frequency of pharynx pumping. The pharyngeal pump frequency was measured after feeding *C. elegans* with the HYS and Δ*hemN2* strains for 24 h. We found no significant difference in the posterior pharyngeal pump rate between the HYS strain (164 ± 12.49 times/min) and Δ*hemN2* (145.2 ± 18.9 times/min). This indicated that *hemN2* deletion did not significantly reduce the feeding rate of *C. elegans* to the HYS strain (Figure 2B). The first and crucial step for a pathogen to exert its virulence is to colonize the host. The *hemN2* knockout strain significantly reduced its colonizing ability in the gut of *C. elegans* (Figure 2C). These results indicated that *hemN2* participated in bacterial virulence by influencing the ability of bacteria to colonize the host gut.

### 3.3. HemN2 Is Regulated Transcriptionally by GacS and Affects 7-HT Production

The Gac system, Cbr/Crc system, pantothenic acid synthesis, and fluorescent pyoverdine pathway are the key pathways for *P. donghuensis* HYS pathogenicity. To further investigate the correlation between *hemN*-related genes and the four virulence pathways in HYS strains, mutant strains of key genes of the four virulence pathways (Δ*gacS*, Δ*cbrA*, Δ*panB*, and Δ*pvdA*) were applied to test the relative expression of four *hemN* genes. The expression of *hemN1* and *hemN2* in Δ*gacS* increased 7.2-fold and 5.0-fold compared to that in the HYS strain, respectively. In contrast, the expression of *hemN4* decreased 3.6-fold (Figure 3A), indicating that the absence of a Gac system significantly promoted the expression of *hemN1* and *hemN2*, but repressed the expression of *hemN4*. The Gac system is a global regulatory system that controls various virulence factors in bacteria. Although *hemN2* was negatively regulated by *gacS*, it still contributed to the virulence effect of *P. donghuensis* HYS to *C. elegans*. This suggested that *hemN2* was not the primary factor in the virulence of *gacS*.

*P. donghuensis* HYS secretes two types of iron scavengers: pyoverdine and nonfluorescent siderophore 7-HT. 7-HT is the virulence factor of *P. donghuensis* HYS. The production of siderophore can be judged by its characteristic absorption peaks, which are at 330 nm and 392 nm for 7-HT and at 405 nm for pyoverdine. At the characteristic peak of 7-HT at 330 nm (Figure 3B), the absorbance peaks of Δ*gacS* (*p* < 0.0001) and Δ*hemN2* (*p* < 0.0001) were significantly reduced compared to the HYS strain (Figure 3C). The results revealed that *hemN2* was regulated by *gacS* and affected the virulence of *P. donghuensis* HYS by influencing the production of 7-HT.

### 3.4. HemN2 Is Involved in Bacterial Virulence through Redox

The T-AOC levels of Δ*gacS* (3.258 ± 0.224 mM) and Δ*hemN2* (3.186 ± 0.108 mM) were significantly higher than the wild-type HYS (2.965 ± 0.123 mM) (Figure 4A). The nitrite levels in Δ*gacS* (0.614 ± 0.107 μM) and Δ*hemN2* (3.805 ± 0.168 μM) were significantly lower than the wild-type HYS (5.211 ± 0.3583 μM) (Figure 4B). The nitrate levels in Δ*gacS* (0.908 ± 0.428 μM) and Δ*hemN2* (5.325 ± 0.466 μM) were significantly decreased compared to the wild-type HYS (8.414 ± 0.408 μM) (Figure 4C). Furthermore, the T-AOC, nitrite, and nitrate levels in the complemented strain Δ*hemN2*/pBBR2-*hemN2* could be restored to the wild-type HYS level. These results suggested that *gacS* and *hemN2* deficiency led to increased total antioxidant levels and decreased oxidative stress capacity, thereby reducing bacterial virulence.

### 3.5. HemN2 Is Involved in the Pathogenicity of P. donghuensis HYS to Zebrafish

To further investigate the role of HemN2 in the pathogenicity of *P. donghuensis* HYS towards other hosts, we compared the amino acid sequence similarity of HemN2 in *P. donghuensis* HYS, *P. aeruginosa*, *P. putida*, *P. fluorescens*, and *P. syringae* and the result was 70.82% (Appendix A). We further constructed phylogenetic trees according to the amino acid sequences of HemN2 in pathogenic *Pseudomonas*. The results showed that *P. donghuensis* HYS was closest to the fish pathogen *P. putida* (Figure 5A). Since *P. donghuensis* HYS was isolated from the waters of the East Lake (China), it is speculated that it may be pathogenic to fish. Therefore, we tested the pathogenicity of wild-type HYS and its derived strains to zebrafish. The zebrafish were infected with wild-type HYS and Δ*hemN2* bacterial culture by immersion, respectively. When the concentration of bacterial suspension was 6 × 10^6^ CFU/mL, the survival rate of zebrafish infected with the Δ*hemN2* strain (LT_50_ 18.0 ± 0.599 h) was significantly higher than the wild-type HYS (LT_50_ 9.6 ± 1.074 h) (Figure 5B,C). No deaths occurred in the negative control group. The results showed that *hemN2* deletion significantly reduced the pathogenicity of *P. donghuensis* HYS to zebrafish.

## 4. Discussion

HemN has been characterized in *E. coli* [42], *Salmonella typhimurium* [43], *P. aeruginosa* [44], *Ralstonia eutropha* [45] and *Bradyrhizobium japonicum* [15]. Former functional studies of HemN had focused on the synthesis of heme under anaerobic conditions, but its toxicological function had not been reported yet. There are four *hemN* genes in *P. donghuensis* HYS: *hemN1*, *hemN2*, *hemN3*, and *hemN4*; each of the four *hemN* genes encodes anaerobic coproporphyrinogen III oxidase, but the sequence similarity among them is only 43.94% (Appendix A). In this study, slow-killing assays showed that the LT_50_ value of *C. elegans* exposed to Δ*hemN2* was prolonged by 1.7 times compared with wild-type HYS, indicating that Δ*hemN2* significantly reduced its virulence. However, the other three genes did not significantly reduce virulence (Figure 1A,B). The restoration of virulence in the complemented strain Δ*hemN2*/pBBR2-*hemN2* to that in the wild-type strain further confirmed the role of *hemN2* for bacterial virulence (Figure 1E,F). Additionally, *C. elegans* exposed to Δ*hemN12* and Δ*hemN124* exhibited a similar trend of reduced virulence as observed with Δ*hemN2*, whereas Δ*hemN124* had a more pronounced effect than Δ*hemN12* (Figure 1C,D). The results of slow-killing assays showed that the *hemN2* gene was the primary contributor to the virulence of *P. donghuensis* HYS, and *hemN4* may serve as compensation in the absence of both *hemN1* and *hemN2*. Combined with the above results, the results indicated functional difference among the four *hemN* genes. A similar phenomenon was reported in a gram-negative soil bacterium *Bradyrhizobium japonicum*, which contained *hemN1* and *hemN2* with 53% amino acid sequence identity. The *hemN2* gene was involved in symbiotic nitrogen fixation with the soybean host plant under anaerobic conditions, whereas *hemN1* did not have this function [15]. The role of hemoglobin in the virulence of bacteria has been documented [46]. Hemoglobin in *Porphyromonas gingivalis* associated with periodontitis influences strain virulence in a concentration-dependent manner [47], and *hmuY* is a significant virulence factor for the infection of macrophages in a hemoglobin-restricted host environment [48].

The initial and crucial stage of pathogenicity is the successful colonization of pathogenic bacteria within the host [49]. *C. elegans* growth is sustained by constant feeding, which is dependent on pharyngeal pumping frequency [50]. Through measurement of the colonization ability of bacteria and the feeding rate of *C. elegans*, our results showed that *hemN2* deficiency did not affect the feeding status of *C. elegans* (Figure 2B). However, it significantly reduced the colonization of *P. donghuensis* HYS in the *C. elegans* intestine (Figure 2C). These results suggested that *hemN2* participated in bacterial virulence via regulation of *P. donghuensis* HYS to colonize the *C. elegans* gut.

The Gac system, Cbr/Crc system, pantothenic acid synthesis, and fluorescent pyoverdine pathway are key pathways through which HYS exerts its pathogenicity [7,10,12]. We further investigated the relationship between *hemN* genes and these four virulence pathways in *P. donghuensis* HYS. It was demonstrated that *hemN* genes were regulated by the Gac system. The GacS/GacA is a global regulatory system that primarily controls bacterial pathogenicity by regulating virulence factors [12]. *gacS* or *gacA* mutants can cause partial or total loss of the biocontrol ability of plant-beneficial *Pseudomonas* and significantly reduce the virulence of pathogenic *Pseudomonas* to plants or animals [51]. For *P. aeruginosa*, the Gac system positively regulates the production of virulence factors such as pyocyanin [52], cyanide, lipase and elastase [53,54,55,56], and is also related to the type III and IV secretion systems and infection function [57,58]. In *P. fluorescens* F113, the Gac system negatively regulates flagellar motility and positively regulates biofilm formation [59]. In *P. fluorescens* CHA0, mutations in the *gacS* and *gacA* genes lead to increased bacterial motility [60]. The study showed that the Gac system is also involved in the synthesis of extracellular polysaccharides, alginate and syringotoxin from *P. syringae* [61]. In conclusion, many phenotypes of pathogenic bacteria are dependent on the Gac system.

The absence of *gacS* significantly promoted the expression of *hemN1* and *hemN2*, but repressed the expression of *hemN4* (Figure 3A). It indicated that the Gac system in HYS has a significant negative regulatory effect on *hemN1* and *hemN2*; meanwhile, it had a significant positive regulatory effect on *hemN4*. Δ*hemN2* showed significantly lower virulence, indicating that although *hemN2* was regulated by the Gac system, it might not be the main factor in the virulence of the Gac system. The Gac system, a *Pseudomonas* global toxicity control system, may be critical in *P. donghuensis* HYS virulence. Previous reports have indicated that GacS/GacA regulated small RNA (RsmY/RsmZ) in *P. donghuensis* HYS, which bound to post-transcriptional repressors (RsmA/RsmE) to deregulate the repressive effects on their possible target genes (*orf1* and *orf12*), and *orf12* stimulated the expression of *orf6*-*orf9*, which regulated the production of 7-HT through an unknown synthetic pathway [12,62]. In *P. aeruginosa,* the GacS/GacA two-component regulatory system, in which the sense kinase GacS and two other sense kinases, LadS and RetS, regulated the synthesis of RsmZ and RsmY by receiving different signals, it has been shown that LadS is associated with the type III secretion system of bacteria, whereas RetS was associated with the synthesis of some pathogenic factors [63]. The regulatory relationship between the Gac system and heme-related genes such as *hemN* is worthy of further investigation and discussion.

*P. donghuensis* HYS high-yield non-fluorescent iron scavenger 7-HT is a secondary substance involved in the virulence of HYS. Several mechanisms had been proposed for the synthesis of 7-HT [10,11]. Quantification of 7-HT siderophore production indicated a significant decrease in the 7-HT producing capacity of Δ*hemN2* compared to the wild-type HYS. The ∆*hemN2*/pBBR2-*hemN2* complemented strain exhibited a phenotype similar to the wild-type HYS strain, which restored the production of 7-HT compared with the ∆*hemN2* strain (Figure 3B,C). It meant HemN2 participated in bacterial virulence by affecting the production of 7-HT in *P. donghuensis* HYS.

The role of oxidative stress in the virulence of *P. aeruginosa* had been reported in the literature. Nitrite transporters have been shown to be necessary for *P. aeruginosa* to form a biofilm defense against the host because they regulate •NO levels [64]. Intestinal bacterial pathogens have been shown to use nitrate to cause intestinal infection [65]. In *Salmonella typhimurium*, nitrate has been shown to inhibit the production of curli components of the biofilm, activate the flagellum via intracellular levels of cyclic di-GMP, and regulate the bacterial sessile-to-motile lifestyle transition, which played a central role in adaptation during infection [66]. The Gac system has been shown to mediate the redox level of bacteria in vivo, and GacS activity in *P. aeruginosa* to regulate the expression of virulence genes associated with acute or chronic infections through transcriptional and post-transcriptional mechanisms [67]. GacA in *P. savastanoi* has been shown to regulate antioxidative stress genes [68]. Compared to wild-type HYS, the knockout strains Δ*gacS* and Δ*hemN2* exhibited a significant increase in antioxidant levels and a significant decrease in nitrite and nitrate levels (Figure 4). HemN2 and GacS in *P. donghuensis* HYS were involved in oxidative stress processes, which led to redox disturbances through ROS and RNS metabolic pathways and participated in the virulence to *C. elegans* (Figure 6). These findings will help build a deeper understanding of interactions between bacteria and hosts, as well as their toxicity.

To further investigate the role of HemN in the pathogenicity of *P. donghuensis* HYS towards other hosts, we constructed a phylogenetic tree using the amino acid sequence of HemN2 from pathogenic *Pseudomonas*. We found that HYS was closely related to the fish pathogen *P. putida*. Furthermore, since HYS was isolated from Donghu Lake in China, we hypothesized that HemN might be involved in the pathogenicity of HYS towards vertebrate model zebrafish. To test this hypothesis, we evaluated the pathogenicity of HYS and its relevant mutant strains against zebrafish. The results showed that the virulence of the HemN2-deficient strain was significantly reduced upon infection of zebrafish (Figure 5). These results suggest that HemN2 plays a critical role in the pathogenicity of HYS towards zebrafish. This suggests that the virulence pathway of HYS may be conserved in *C. elegans* and zebrafish. However, the immune systems of the two species are different, and whether the virulence mechanism of HemN2 in HYS towards *C. elegans* is the same as that towards zebrafish needs to be further investigated. In addition, our studies have mainly focused on the regulation of virulence in HYS. The possible physiological and biochemical responses of HYS to the host deserve further systematic study.

Altogether, HemN2 was regulated by the Gac system and was involved in bacterial virulence towards *C. elegans* by influencing 7-HT synthesis and oxidative stress. The regulatory relationship between HemN2 and the Gac system and the association of HemN2 with key genes involved in 7-HT synthesis and oxidative stress remain to be determined. Therefore, future research should investigate how HemN2 regulates genes associated with redox reactions and 7-HT synthesis, thereby mediating bacterial virulence. In addition, the relationship between HemN2 and the known virulence pathways of *P. donghuensis* HYS deserves further investigation. Finally, considering the pathogenicity of HemN2 in *P. donghuensis* HYS towards the model organisms *C. elegans* and zebrafish, these insights provide further understanding of HemN-based virulence in the genus *Pseudomonas*, and this is expected to provide a new drug target for fish disease control and reveals the importance of *Pseudomonas* in aquatic animal diseases.

## 5. Conclusions

*P. donghuensis* HYS exhibits stronger virulence towards *C. elegans* compared to the pathogenic *P. aeruginosa* and *P. putida*. Previous research on HemN had primarily focused on its role in heme synthesis, with its correlation to bacterial virulence still unclear. Here we reported for the first time that the deletion of *hemN2* significantly reduced *P. donghuensis* HYS virulence to *C. elegans*, as well as its colonization in the *C. elegans* gut. Similarly, this deletion weakened *P. donghuensis* HYS pathogenicity in zebrafish larvae. Further studies showed that *hemN2* was regulated by the GacS system and affected bacterial virulence by influencing the synthesis of the virulence factor 7-HT, antioxidant capacity and nitrification levels. This discovery should help researchers better understand distinctive pathogenicities of the *Pseudomonas* genus and provide new perspectives for *Pseudomonas* for the prevention and treatment of aquatic animal diseases.

## Figures and Tables

**Figure 1 biology-13-00373-f001:**
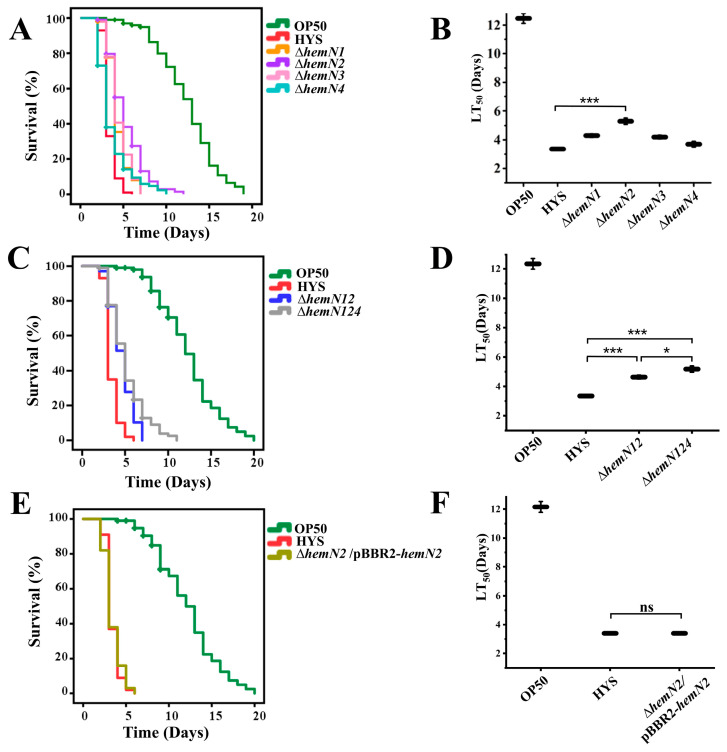
Involvement of *hemN2* in the pathogenicity of HYS strain was observed in a slow-killing model of *C. elegans*. (**A**,**B**) Mutants of *hemN1*, *hemN2*, *hemN3*, and *hemN4* were tested using slow-killing experiments. (**C**,**D**) Virulence of the double-knockout mutant Δ*hemN12* and triple-knockout mutant Δ*hemN124*. (**E**,**F**) Genetically complemented strain Δ*hemN2*/pBBR2-*hemN2* further confirmed the virulence function of *hemN2*. Data represent the mean ± SD, *n* = 100. * *p* < 0.05, *** *p* < 0.001, Student’s *t*-test.

**Figure 2 biology-13-00373-f002:**
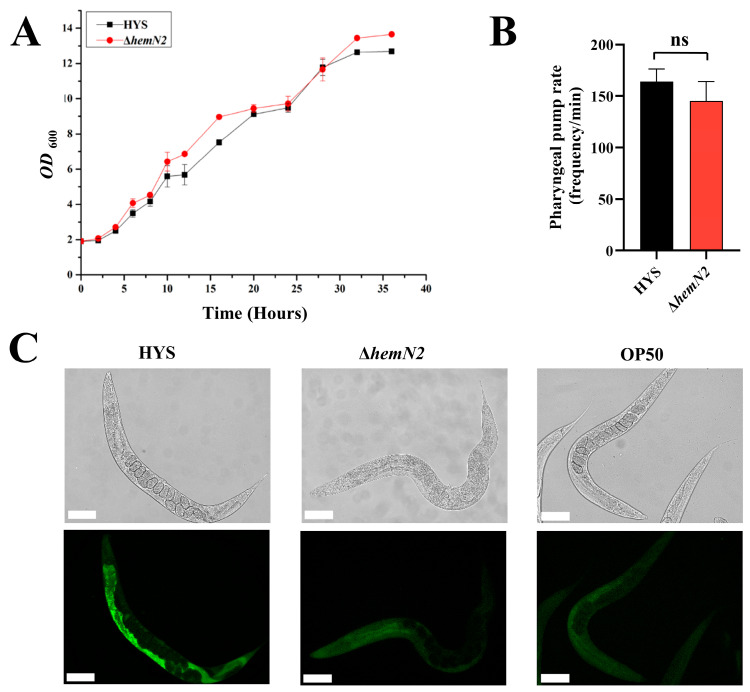
*hemN2* regulated by *gacS* is involved in the pathogenicity of *P. donghuensis* HYS to *C. elegans*. (**A**) Growth curve of the Δ*hemN2* strain and HYS. Growth curves of the strains were measured on NGM plates. (**B**) Number of pharyngeal pumps per minute of *C. elegans* was counted under a microscope. (**C**) Intestinal colonization by HYS strain, Δ*hemN2*, and OP50 strains to *C. elegans*. Scale bars = 100 μm. Data represent the mean ± SD, *n* = 10. Student’s *t*-test.

**Figure 3 biology-13-00373-f003:**
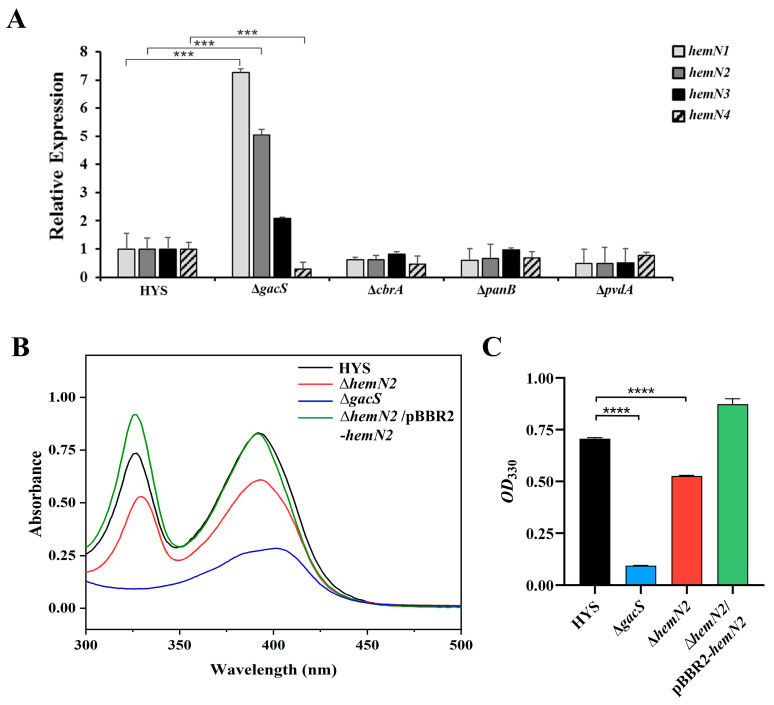
In *P. donghuensis* HYS, HemN was regulated by GacS and affected the production of the virulence factor 7-HT. (**A**) Transcription levels of *hemN*-related genes were detected by the Gac system (Δ*gacS*), Cbr/Crc system (Δ*cbrA*), pantothenic acid (Δ*panB*), and fluorescent pyoverdine pathway (Δ*pvdA*) key gene deletion strains. (**B**) Absorption spectra of supernatants of HYS, Δ*gacS*, and Δ*hemN2* after 24 h of incubation in MKB liquid medium. (**C**) Detected production of 7-HT in the culture supernatants of HYS, Δ*gacS*, and Δ*hemN2* in MKB medium at *OD*_330_ nm. Data indicated the mean ± SD from three independent experiments. *** *p* < 0.001, **** *p* < 0.0001, Student’s *t*-test.

**Figure 4 biology-13-00373-f004:**
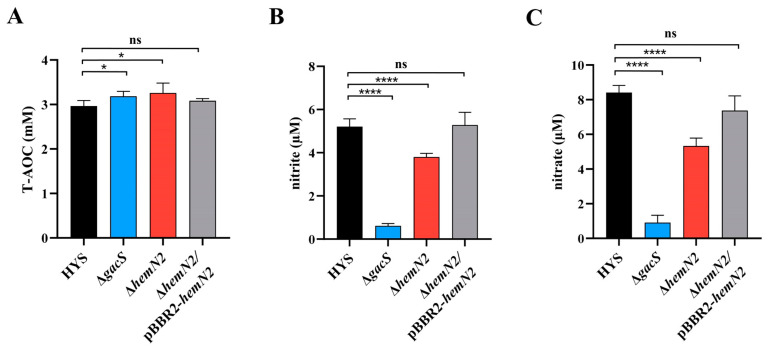
Production redox molecular levels of wild-type HYS, Δ*gacS* and Δ*hemN2* strains. (**A**) Quantification of total antioxidant capacity (T-AOC). (**B**) Quantification of nitrite. (**C**) Quantification of nitrate. Data indicated the mean ± SD from three independent experiments. * *p* < 0.05, **** *p* < 0.0001, Student’s *t*-test.

**Figure 5 biology-13-00373-f005:**
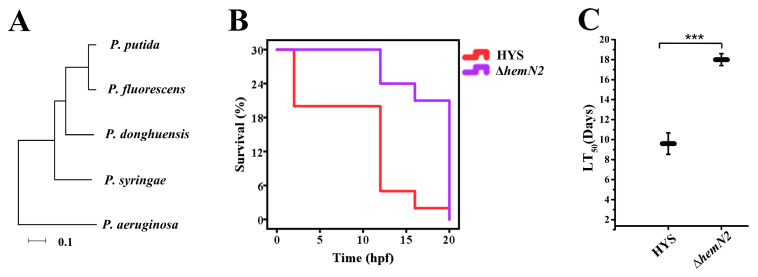
HemN2 is involved in the pathogenicity of *P. donghuensis* HYS to zebrafish. (**A**) Phylogenetic evolutionary tree of *hmeN2* homologues using the neighbor-joining method with a bootstrap value of 1000 using the MEGA7.0 program. (**B**,**C**) Survival rate of zebrafish was measured every 2 h. Data represent the mean ± SD, *n* = 30. *** *p* < 0.001. Student’s *t*-test.

**Figure 6 biology-13-00373-f006:**
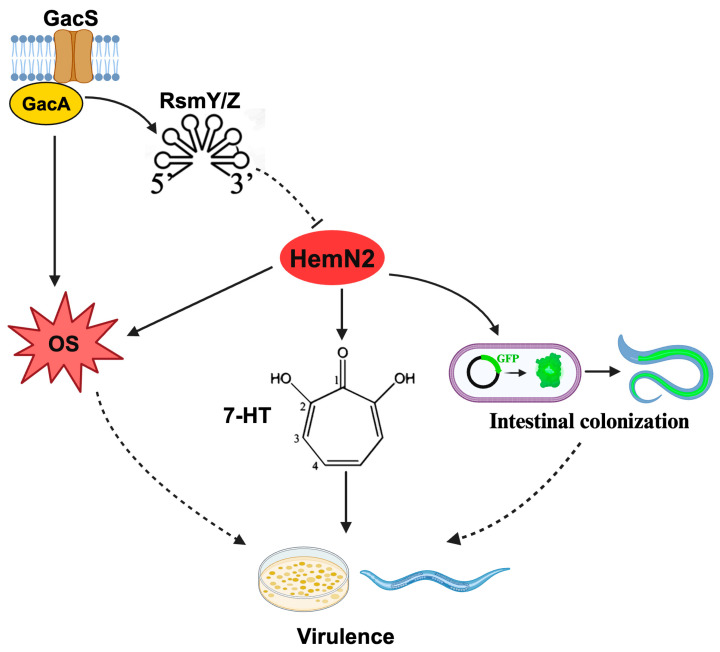
Schematic overview of the regulatory network of HemN2 involved in *P. donghuensis* HYS on host virulence. T-shaped lines indicate the negative control, arrows represent the positive control, solid lines highlight confirmed regulations, and dotted lines indicate unconfirmed regulations in this work. GacS and GacA represent the GacS and GacA proteins of the Gac system. RsmY/Z represent the small RNAs RsmY and RsmZ of the Gac system. 7-HT represents virulence factor 7-hydroxytropolone. OS represents the level of oxidative stress in bacteria.

## Data Availability

Data are contained within the article and Appendix A.

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
