# Peer review of "HemN2 Regulates the Virulence of Pseudomonas donghuensis HYS through 7-Hydroxytropolone Synthesis and Oxidative Stress"

_biology, 2024, doi:10.3390/biology13060373_

Round 1
Reviewer 1 Report
Comments and Suggestions for Authors
Could the author provide further insight into the selection of E. coli OP50 as the control strain? It seems it was utilized as a negative control; as a first-time reader, a succinct explanation about this strain would enhance clarity for the audience.
Why was antioxidant capacity measured instead of oxidative capacity, considering the Gac system is related to NO and ROS?
The methodology regarding 7-HT needs to be elaborated in detail; there are omissions from the original article that should be addressed.
What functional distinctions exist between hemN1-4?
It would be intriguing to explore the pathogenicity of ∆gacS given the significant upregulation of hemN.
Does the Gac system specifically regulate hemN? If not, what other candidates are also under the regulation of Gac?
Author Response
We provide a point-by-point response. Please see the attachment "Response to reviewer1 ".

Reviewer 2 Report
Comments and Suggestions for Authors
Dear Author,
Submitted manuscript "Biology-3009666 titled as- HemN2 regulates the virulence of Pseudomonas donghuensis HYS through 7-hydroxytropolone synthesis and oxidative stress" is a well written and documented manuscript. Here researchers have shown the deletion of hemN2 significantly reduces P. donghuensis HYS toxicity to C. elegans, as well as its colonization in the C. elegans gut. This is an engaging article with robust methodology that purposefully questions our knowledge of the subject. Present form of manuscript has well-presented and understandable to a specialist readership.
The introduction provides sufficient background, and the other sections include results has been clearly discussed and analyzed exhaustively. Paper has cited good references in support although reference number 15 and 16 are very old. The recent article would be better like PMID: 30273394; doi: 10.4172/2168-9652.1000182. This paper has potential to contribute and impact to the field.
All the best.
Comments on the Quality of English LanguageMinor spelling checks and English correction is needed.
Author Response
We provide a point-by-point response. Please see the attachment "Response to reviewer2 ".

Reviewer 3 Report
Comments and Suggestions for Authors
Check some words and sentences with grammar errors
Author Response
We provide a point-by-point response. Please see the attachment "Response to reviewer3 ".
